# Evidence for hybrid breakdown in production of red carotenoids in the marine invertebrate *Tigriopus californicus*

**Matthew J. Powers**[1☯]*, **Lucas D. Martz**[2☯]*, **Ronald S. Burton**[2‡], **Geoffrey E. Hill**[1‡], **Ryan J. Weaver**[3‡]

**1** Department of Biological Sciences, Auburn University, Auburn, AL, United States of America, **2** University of California, Scripps Institution of Oceanography, San Diego, CA, United States of America, **3** Ecology, Evolution, and Organismal Biology Department, Iowa State University, Ames, IA, United States of America

☯ These authors contributed equally to this work.
‡ These authors also contributed equally to this work.
* mjp0044@auburn.edu (MJP); ldmartz97@gmail.com (LDM)

**Data Availability Statement:** All relevant data are within the manuscript and its Supporting Information files.

## Abstract

The marine copepod, *Tigriopus californicus*, produces the red carotenoid pigment astaxanthin from yellow dietary precursors. This 'bioconversion' of yellow carotenoids to red is hypothesized to be linked to individual condition, possibly through shared metabolic pathways with mitochondrial oxidative phosphorylation. Experimental inter-population crosses of lab-reared *T. californicus* typically produces low-fitness hybrids is due in large part to the disruption of coadapted sets nuclear and mitochondrial genes within the parental populations. These hybrid incompatibilities can increase variability in life history traits and energy production among hybrid lines. Here, we tested if production of astaxanthin was compromised in hybrid copepods and if it was linked to mitochondrial metabolism and offspring development. We observed no clear mitonuclear dysfunction in hybrids fed a limited, carotenoid-deficient diet of nutritional yeast. However, when yellow carotenoids were restored to their diet, hybrid lines produced less astaxanthin than parental lines. We observed that lines fed a yeast diet produced less ATP and had slower offspring development compared to lines fed a more complete diet of algae, suggesting the yeast-only diet may have obscured effects of mitonuclear dysfunction. Astaxanthin production was not significantly associated with development among lines fed a yeast diet but was negatively related to development in early generation hybrids fed an algal diet. In lines fed yeast, astaxanthin was negatively related to ATP synthesis, but in lines fed algae, the relationship was reversed. Although the effects of the yeast diet may have obscured evidence of hybrid dysfunction, these results suggest that astaxanthin bioconversion may still be related to mitochondrial performance and reproductive success.

**Funding:** Funding for this project received by GEH and RSB. Grant/Award from National Science Foundation, Division of Integrative Organismal Systems, NSF-IOS 1701827. https://www.nsf.gov/div/index.jsp?div=IOS. The funders had no role in study design, data collection and analysis, decision to publish, or preparation of the manuscript.

**Competing interests:** The authors have declared that no competing interests exist.

## Introduction

Carotenoids pigments are widely distributed across diverse groups of organisms where they serve critical roles in physiological processes. Despite their utility, carotenoids are produced only by plants, fungi, algae, and bacteria. With a few notable exceptions [1], animals lack the biochemical pathway necessary to synthesize carotenoids and therefore must obtain carotenoids from their diet to support key functions within their bodies and to produce colorful external displays [2]. Once ingested and absorbed, carotenoids may then be metabolized into new forms that perform distinct functions such as vitamin-A synthesis, pro-oxidant defense, and others [3–8]. Some animal taxa have co-opted carotenoids for their capacity to reflect yellow to red wavelengths of light and use them in colorful external displays [2]. The identity of carotenoids used by many animals for coloration have been identified using chromatography and spectrophotometric techniques. This foundational research has identified carotenoid pigments in the integuments of some animals that are not typically found in their diet, suggesting that they are metabolizing or 'bioconverting' carotenoids from their diet. For example, yellow dietary carotenoids, such as zeaxanthin or β-carotene, can be hydroxylated or ketolated into the red ketocarotenoid astaxanthin [9, 10], and astaxanthin, in turn, may perform dual roles as a vibrant colorant and potent protector against oxidative stress, particularly in marine animals [11–13].

The observation that animals bioconvert dietary carotenoids to new forms used for coloration begs the question as to what genes are involved and what is the intracellular site of the metabolic pathways? Recently, *CYP2J19* was identified as the gene that encodes a putative β-carotene ketolase responsible for the bioconversion of yellow carotenoids to astaxanthin is birds and turtles [14, 15]. Other taxa including frogs [16], mites [17], and crustaceans [18] have provided some candidate genes for the yellow to red carotenoid bioconversion, but more work is required. The exact intracellular arena for carotenoid bioconversion within tissues, however, has yet to be determined [19–23]. Recent hypotheses and molecular modeling place the cellular site of carotenoid bioconversion within the mitochondria or in association with mitochondrial associated membranes, implicating a shared metabolic pathway between keto-carotenoid conversion and cellular respiration [22, 24–26].

The idea that carotenoid metabolism is linked to cellular respiration has been supported by observations in diverse species of birds. Ketocarotenoids were found in high concentrations in the liver mitochondria of house finches (*Haemorhous mexicanus*) [27]. Within the mitochondria of this bird, ketocarotenoids were localized in the highest concentrations at the inner mitochondrial membrane [26]. Preliminary models predict CYP2J19 should localize either in or around the mitochondria [26, 28, 29]. Moreover, the concentration of ketocarotenoids found in the feathers of house finches correlated strongly with the ability of their mitochondria to respond to changes in respiration and to withstand greater cumulative levels of mitochondrial stress [26]. Correlations between hormonal signaling, longevity, and pigmentation have also been observed in red-legged partridges (*Alectoris rufa*), with mitochondrial function implicated as the underlying causative factor [30]. Along with positive associations between mitochondrial function and production of red pigments, experiments with zebra finches [31] and red crossbills [32] revealed an effect on ketocarotenoid bioconversion by redox-active compounds targeted to the inner mitochondrial membrane. However, to better understand the link between mitochondrial respiration and carotenoid ketolation, observations in taxa outside of Aves are needed.

Bioconversion and accumulation of ketocarotenoids is a prominent feature of many crustaceans. In this often-colorful group, red and blue pigmentation is frequently produced by the accumulation of the ketocarotenoid astaxanthin, along with other ketocarotenoids [13].

Notably, the precursors and products in ketocarotenoid pathways in many crustaceans are the same as those in birds, and the oxidizing enzymes involved in these crustacean pathways are predicted to be functionally similar to the enzymes employed by avian species [18, 33, 34]. One particular crustacean group, oceanic and lake-dwelling copepods, have frequently been utilized in carotenoid pigmentation studies due to their critical link in ecological food webs and their amenability to experimental manipulation [8, 35, 36]. Marine copepods from the genus *Tigriopus* have been particularly useful to investigate the condition-dependency of carotenoid pigmentation [12, 37] and the dynamics or pathways of carotenoid bioconversion [9, 34]. Previous work has established that the major carotenoid accumulated in the tissues of *T. californicus* is free astaxanthin, compared to much lower concentrations of esterified astaxanthin and dietary carotenoid precursors [9, 34]. However, *Tigriopus californicus* copepods are perhaps best known as a model for studies of mitonuclear coadaptation [38, 39].

Interpopulation crosses of *T. californicus* show highly variable and often reduced mitochondrial function and reproductive fitness [40–42]. These negative effects of hybridization have been traced to incompatibilities between the encoded products of one population's nuclear genome, which consists of 12 chromosomes [38], and the products of the non-coadapted mitochondrial genome of the other population [39, 43]. Nuclear and mitochondrial genes within populations co-evolve to maintain key interactions between proteins that make up Complexes I, III, IV and V of the electron transport system (ETS) [44, 45]. In *T. californicus*, hybridization between genetically divergent populations breaks up co-evolved combinations of nuclear and mitochondrial genes thanks to maternal inheritance of mt-DNA and sexual recombination of nuclear DNA. The end result in terms of mitochondrial performance and organismal fitness is variable in the second generation of hybrids and beyond (once hybrid offspring no longer retain at least one full copy of maternal genes) [41]. Some hybrid lines show decreased function in all ETS protein complexes (with the exception of Complex II that is entirely nuclear encoded), increased oxidative stress, and reduced fecundity [40, 42, 43, 46, 47].

In this study, we utilized interpopulation hybrids of *Tigriopus californicus* that have previously shown mitochondrial dysfunction from mitonuclear incompatibilities [39, 40, 42, 43, 48]. We tested the hypothesis that the efficiency of carotenoid ketolation is correlated with mitochondrial function and predicted that mitonuclear mismatched hybrid lines would produce less astaxanthin than corresponding non-hybrid, parental lines. We also assessed the relationship between astaxanthin production and measures of mitochondrial performance and fitness among all lines. Establishing links between carotenoid metabolism, mitochondrial function, and fitness measures have outstanding potential to better understanding the evolution of this conspicuous coloration in animals.

## Materials and methods

### Copepod sampling and culturing

*Tigriopus californicus* copepods were collected along the west coast of California and Baja California (Fig 1). Copepods were collected under collection permit #SCP-339 from California Fish and Wildlife. We used copepods sampled from the following locales listed geographically from most southern to most northern: La Bufadora (BUF), San Diego (SD), Bird Rock (BR), Catalina Island (CAT), Abalone Cove (AB), Santa Cruz (SCN) and Pescadero (PES). We reared copepods from each population separately in large beakers containing 35 psu seawater maintained at 20 C on 12h light: dark cycles. We fed copepods used in carotenoid bioconversion assays a diet of ground nutritional yeast (Bragg, Santa Barbara, CA) from birth to produce individuals deficient of carotenoids and clear/white in color. This ground yeast is inactive yeast powder, supplemented with B vitamins, and contains no fats.

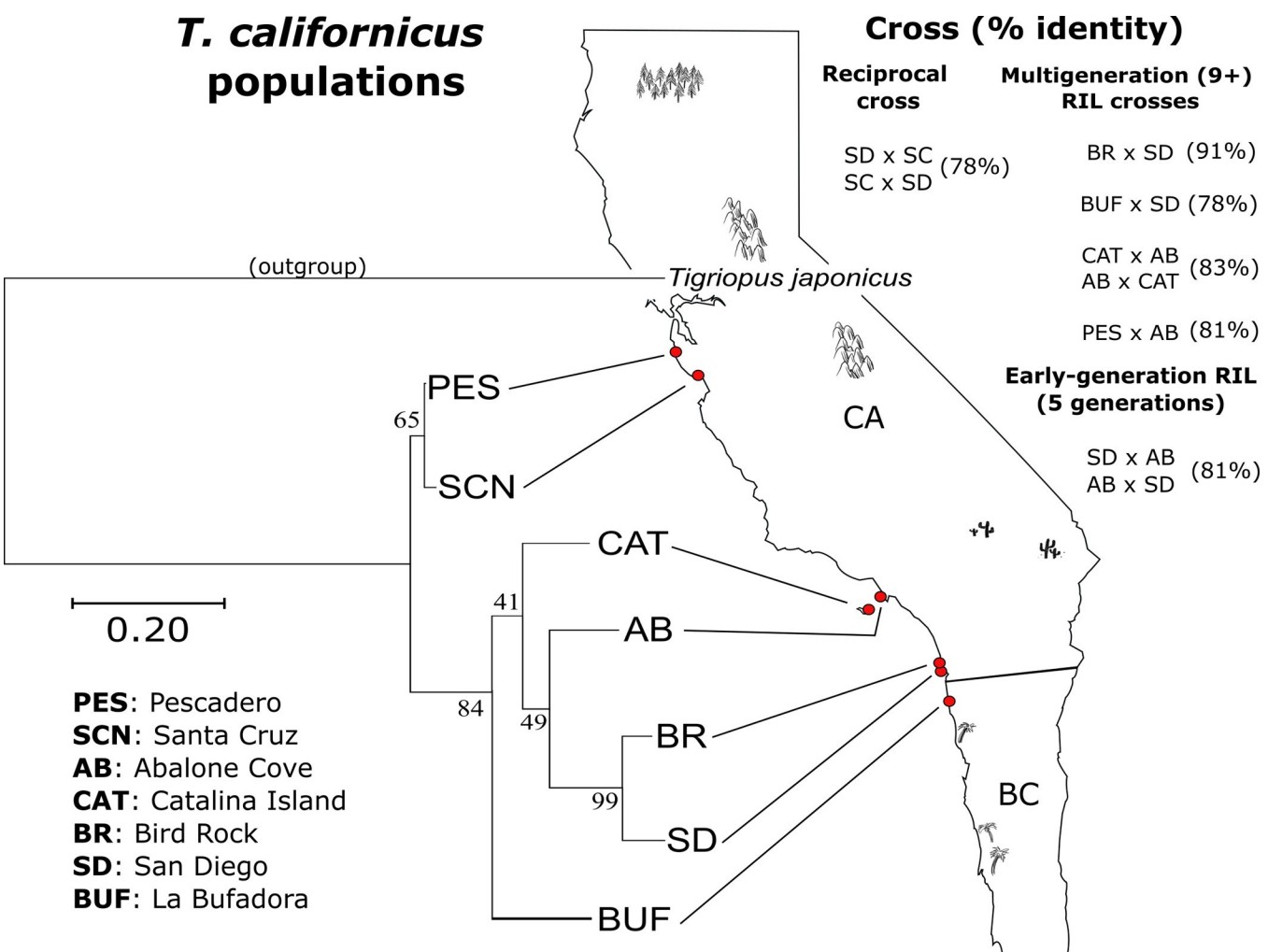

**Fig 1. Sampling locations along the Pacific Ocean stretching from La Bufadora (BUF) in Baja California (BC) to Pescadero (PES) in the north of California (CA).** Percent identity of mitochondrial CO1 gene sequences are shown in parentheses. Populations sampled in this study are shown on the left organized into a phylogenetic tree in Mega X based on 611 bp CO1 sequences, using *Tigriopus japonicus* as a rooted outgroup. For the full methods used to create the tree, see Supplemental Information section 1.6 (S1 File). Key: PES = Pescadaro, SCN = Santa Cruz, AB = Abalone Cove, CAT = Catalina, BR = Bird Rock, SD = San Diego, BUF = La Bufadera.

## Experimental crosses and creation of Recombinant Inbred Lines (RILs)

The methods used to create hybrid copepod lines (see Fig 1 for summary) are described here briefly. For a more detailed description of hybrid line generation, see S1 Table for a summary of all crosses and the supplemental information for expanded methodological details.

Multigeneration RILs were formed by crossing the SD population with either BUF or BR populations (Southern RILs), and by crossing the AB population with the CAT or PES populations (Northern RILs). Southern crosses involved pairing SD males with either BR or BUF females (SD nuclear genes tested against BR or BUF mitochondrial genes), and the Northern crosses involved the pairing of AB males with either CAT or PES females (AB nuclear genes tested against CAT or PES mitochondrial genes) and also one pairing of AB females with CAT males (CAT nuclear genes tested against AB mitochondrial genes). Hybrid matings were performed in duplicate and offspring from one replicate were paired with offspring from another to avoid inbreeding. This was repeated until the F3 generation, at which point iso-female,

inbred lines were established from a single F3 female's clutch. Beginning in the F4, each generation was initiated with a single pair of full sibs. Discrete generations were maintained until the seventh generation, at which point offspring were allowed to mate continuously and were maintained with overlapping generations. Discrete generations were maintained through the seventh generation in order to allow inbreeding among siblings and purge variation in alleles so that each line more closely represented a unique mitonuclear genotype after the effects of recombination in the first three non-inbred generations. This entire process was repeated using males and females from the same population to make PILs (parental inbred lines) as controls for comparison (S1 Table). Similarly, we generated short-term RILs and PILs from a cross of the southern SD and northern AB populations using 20 males and females each in triplicate; however, with this cross we assayed astaxanthin bioconversion at the 5th generation only (S1 Table).

After analyzing mitochondrial data collected from our multigeneration RILs (see Results below), we became concerned that the yeast diet fed to those copepods created a confounding effect of dietary stress on mitochondrial and fitness measurements. Copepods from the BUF and BUFSD19 lines fed only ground yeast produced less ATP than copepods fed a complete algal diet (S1A–S1D Fig). Among all parental lines (controlling for non-independence of data within lines), copepods fed yeast had offspring that developed significantly slower compared to parental lines fed algae (S1E Fig; mean days to copepodid stage ± 95% CI; yeast-only: 7.21 ± 0.41, algae-fed: 6.29 ± 0.73, $t$ = 2.16, $p$ = 0.037). Thus, we performed a fresh reciprocal cross between SD and SCN populations, this time feeding the copepods *Tetraselmis* algae for one week prior to measuring mitochondrial function and during offspring development. The reciprocal cross between SD and SCN populations was formed by pairing 40 males from SD with 40 virgin females from SCN and vice versa in two petri dishes. F1 mating pairs were transferred to a new dish to produce F2 offspring. This process was repeated to produce F3 offspring which were then subjected to fitness and carotenoid measurements (described below).

## Carotenoid bioconversion assays

Prior to the start of the experiment, RIL copepods were switched to a diet of ground nutritional yeast until they became clear in color and deficient of both carotenoid precursors and the primary red carotenoid, astaxanthin [9, 49]. To test carotenoid bioconversion rate, carotenoid-deficient copepods were provided *Tetraselmis chuii* algae *ad libitum* for 7 days to mimic the copepod's natural diet of photosynthetic algae rich with carotenoid precursors [50]. *T. chuii* algae produces multiple dietary carotenoid precursors that can be converted into astaxanthin [9, 34, 51]. On the seventh day, copepods were moved to clean artificial saltwater for a minimum of 2 hours to clear any algae remaining in their gut. Copepods were then dried and weighed (±0.001 mg) before being stored in a microcentrifuge tube at -80°C until HPLC analysis (see below).

## Carotenoid extraction and HPLC analysis

Carotenoids were extracted from dried copepod tissues using acetone with sonication and centrifugation to remove cellular debris. Final carotenoid extract was resuspended in 50 μL acetone for HPLC analysis. We separated and quantified copepod carotenoids using HPLC following Weaver et al (2018). Briefly, we injected 10 μL of suspended carotenoid extract on to a Sonoma C18 column (10 μm, 250 x 4.6 mm, ES Technologies, New Jersey, USA) fitted with a C18 guard cartridge. Carotenoids were separated using a Shimadzu Prominence HPLC system with mobile phases A 80:20 methanol: 0.5M ammonium acetate, B 90:10 acetonitrile: water, and C ethyl acetate in a tertiary gradient of 100%A to 100%B over 4 min, then to 80% C: 20% B

over 14 min, back to 100% B over 3 min, and returning to 100% A over 5 min and held for 6 min [9, 52]. We visualized and detected carotenoid absorbance using a Prominence UV/Vis detector set to 450 nm. We identified and quantified carotenoids by comparison to calibration curves of authentic standards that included: astaxanthin, zeaxanthin, β-carotene, lutein, hydro-xyechinenone, and canthaxanthin. We normalized carotenoid concentration by the dry weight of each copepod sample (reported as μg carotenoid per mg copepod tissue).

Due to restrictions related to the SARS-CoV-2 pandemic, we used a different method to measure astaxanthin content of copepods from the reciprocal cross between SD and SCN populations (analyzed mid-year 2020; see Supplemental Information 1.3 in S1 File). Briefly, astaxanthin content of copepods from these crosses (~10 ind. per replicate) was quantified using an Agilent HPLC 1260 Infinity II LC system with the following mobile phases: A) 50:25:25 Methanol: Acetonitrile: 0.25M Aqueous Pyridine and B) 20:60:20 Methanol: Acetonitrile. For full descriptions of this mobile phase linear gradient, as well as the carotenoid extraction methods for this analysis, see Section 1.3 in S1 File.

## Offspring development rate assay

Gravid females (sample sizes in S1 Table) were removed from yeast-fed cultures and placed into filtered seawater in a 6-well plate. These copepods were supplemented with powdered Spirulina (Jade Spirulina, Salt Creek Inc., Salt Lake City, USA) instead of powdered yeast because mortality of gravid females in 6-well plates was high when fed yeast. The Spirulina powder is not the same *Tetraselmis* algae used to resupply copepods with a diet rich in carotenoids. The main pigment in Spirulina are anthocyanins. The Spirulina diet provides enough nutrition to consistently allow reproduction; *T. californicus* females will delay reproduction or cannibalize offspring if they cannot acquire enough quality food [53]. Plates were monitored daily, and whenever an egg sac hatched, the date was recorded, and the female and any unhatched eggs were removed. The freshly hatched offspring were then monitored daily, and the number of individuals that metamorphosed to the copepodid I form (sixth molt, halfway point of development to full maturity) was recorded for each well on each day [54].

For the F2 and F3 generations of the reciprocal SD and SCN crosses, development rate was assayed slightly differently due to time constraints. 50 gravid females were placed in a petri dish with filtered seawater and Spirulina. The following day, all non-gravid females and unhatched egg sacs were moved to a new dish, leaving behind any nauplii that hatched. This was repeated for 5 days, and the hatch date of each dish was recorded. Dishes with freshly hatched offspring were monitored daily, and any animals that metamorphosed to the first copepodid stage were counted and removed so they were not confused with newly molted copepodids the following day.

## ATP synthesis and citrate synthase assays

We measured *in vitro* ATP production from isolated mitochondria following methods from [42, 55]. For each sample, ten males and ten females were homogenized in 800 μL isolation buffer (400 mM sucrose, 100 mM KCl, 6 mM EGTA, 3 mM EDTA, 70 mM HEPES, 1% w/v BSA, pH 7.6), and their mitochondria were isolated by sequential centrifugation. The isolated mitochondrial pellet was resuspended in ~55μL assay buffer (560 mM sucrose, 100mM KCL, 10mM KH$_2$PO$_4$, and 70mM HEPES). Suspended mitochondria (25μL) from each sample were then added to 5μL of either Complex I (CI) substrate (1 mM ADP, 2 mM malate, 10 mM glutamate, and assay buffer) or Complex II (CII) substrate [1 mM ADP, 10 mM succinate, 0.5 μM rotenone (Complex I inhibitor)]. Samples were incubated for 10 minutes at 20°C to allow for ATP synthesis. After the incubation period, 25μL of each sample was added to a 96-well assay

plate with 25μL of CellTitre-Glo (Promega)—which halts ATP synthesis. Luminescence of the samples and ATP standards was measured on the same plate using a Fluoroskan Ascent FL plate reader (Thermo Labysystems).

A citrate synthase (CS) activity assay was performed on the remaining mitochondrial suspension to estimate mitochondrial volume and standardize ATP production per sample following the methods of [56]. Mitochondrial volume (or mitochondrial content) represents the fraction area of cell volume taken up by the mitochondria [57, 58]. CS activity correlates strongly with the fraction of mitochondrial surface area to total cell surface area in the cell [57], thus we use it as a proxy for mitochondrial content or mitochondrial volume [58] when standardizing ATP production [56]. CS activity was determined as follows: a 5μL aliquot of the mitochondrial suspension was added to 50μL of 200mM Tris buffer with 0.2% Triton-X (Sigma), 24μL DI water, 10μL 1mM DTNB (5, 5'-dithiobis (2-nitrobenzoic acid)), and 6μL acetyl coenzyme A (0.3mM). After estimating background activity, 5μL of 10mM oxaloacetic acid was added to each sample and absorbance read at 412nm. The change in absorbance measured over 5 minutes was used to calculate CS activity.

### Statistical analyses

Samples sizes for our measurements on each line are shown in S1 Table. We used mixed-effects linear models and pairwise contrasts corrected for multiple comparisons (using 'emmeans') to compare astaxanthin concentration among hybrid and non-hybrid copepod lines. These models included a random effect of line ID to account for non-independence of data within each line. We repeated this same analysis on dietary carotenoid concentrations and the ratio of astaxanthin to dietary carotenoid concentrations among lines. We also used linear models and paired contrasts corrected for multiple comparisons to analyze differences in ATP production, and offspring development rate among hybrid and non-hybrid copepods lines.

Because we lacked data from female copepods in several crosses, we used male data to analyze the relationships between astaxanthin production and ATP production and offspring development rate among all of the lines using mixed effects models. We included cross ID (i.e., "BR x SD", "SD x SD", etc.) as a random effect to account for non-independence of data from the same hybrid cross or parental control line. Before fitting the models, ATP production and development rate data were log transformed, scaled, and centered to achieve normality of model residuals. We averaged replicates from each line to avoid pseudoreplicaiton and because the same individuals could not be used for all measurements (i.e., the same copepods could not be sacrificed for both astaxanthin extraction and mitochondrial measurements).

All statistical analyses were performed in R. For the full list of packages used, see supplemental info section 1.5 (S1 File).

## Results

### Recombinant inbred lines

Among the multigeneration inbred lines, we found no clear pattern between non-hybrids and hybrids in ATP production (S2 Fig) or offspring development rate (S3 Fig). Two lines were significantly higher than others in ATP production: the CAT parental line (Complex I, S2B Fig) and the BRSD56 hybrid line (Complex II, S2C Fig). Thus, in the multigeneration RILs fed yeast only, we detected no evidence of mitochondrial dysfunction, even in RILs from crosses of highly diverged populations.

However, we did observe differences in astaxanthin production among male copepods from hybrid and non-hybrid lines (Fig 2), but no difference in astaxanthin bioconversion

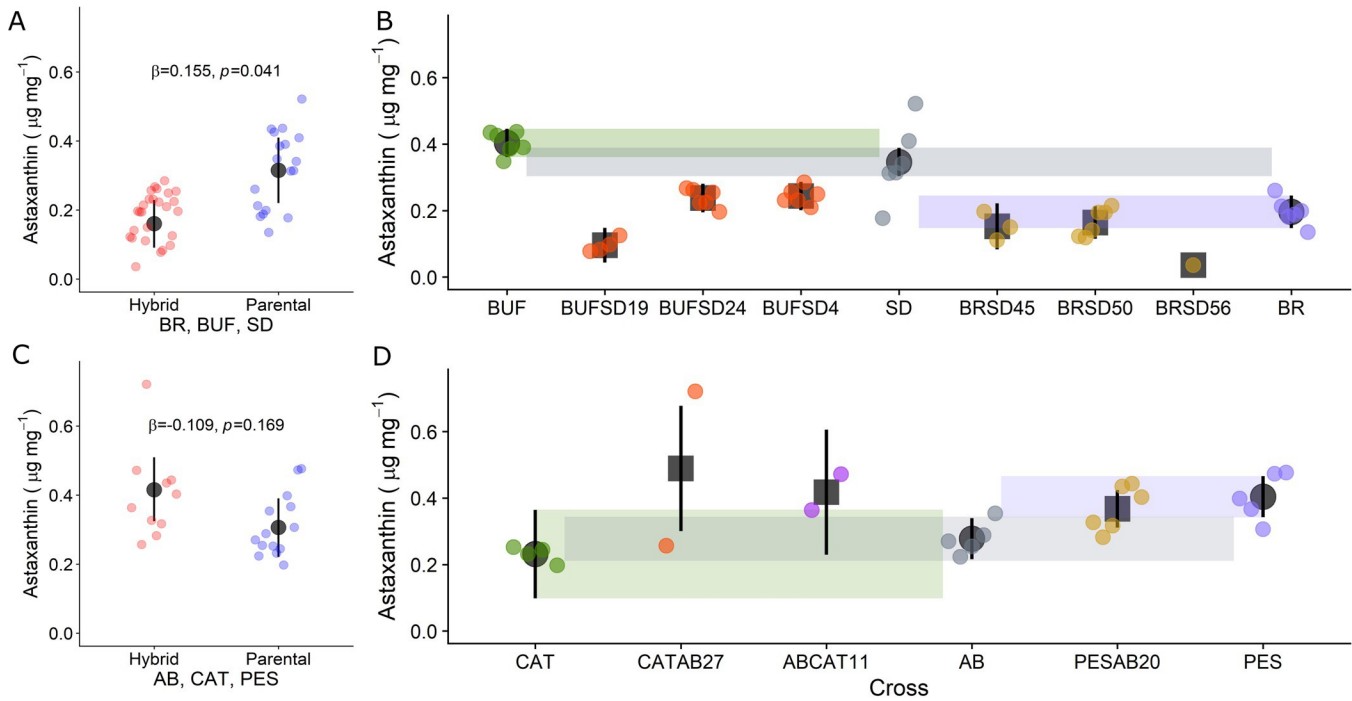

**Fig 2. Differences among hybrid RILs and non-hybrid PILs from multigeneration crosses.** Colored dots represent replicates within each line that are the average astaxanthin concentration of approximately 10 copepods. Dark grey circles (parental lines) and dark grey squares (hybrid lines) are the line average. Black lines and colored shading are 95% confidence intervals around the line average. Where 95% confidence intervals do not overlap, there is a statistically significant difference between lines (S2 Table).

among female copepods from different lines (S4 Fig). Additionally, males produced significantly more astaxanthin than females on average, even while controlling for line ID (mean µg mg$^{-1}$ astaxanthin ± SE; male = 0.27 ± 0.03; female = 0.21 ± 0.03; n = 124; df = 7; $p = 0.001$). Because of this sex effect and the lack of female data in the northern lines (S4B Fig), we analyzed male data only for the rest of the analyses involving the RILs and PILs below.

The difference in astaxanthin production among male copepods across all of the multigenerational RIL and PIL lines was not consistent, nor did it closely match with pairwise genetic divergence (Figs 1 and 2). Hybrid males from crosses of southern populations produced less astaxanthin than parental lines, on average (mean µg mg$^{-1}$ astaxanthin ± SE; hybrid = 1.6e$^{-3}$ ± 3.6e$^{-4}$; parental = 3.2e$^{-3}$ ± 4.9e$^{-4}$; n = 44; df = 7; $p = 0.035$; Fig 2A). Moreover, this difference in astaxanthin production was driven by RILs of BUF and SD hybrids (Fig 2B). RILs from the hybrid crossing of female BUF copepods with male SD copepods (BUFSD19, BUFSD24, BUFSD4) accumulated significantly less astaxanthin compared to both SD and BUF parental lines (Fig 2 and S2 Table). RILs from the crossing of female BR copepods with male SD copepods (BRSD45, BRSD50, BRSD56) accumulated significantly less astaxanthin than the SD parental line but only slightly less than the BR parental line (Fig 2 and S2 Table). Individual RIL lines from the Northern crosses (ABCAT11, CATAB27, and PESAB20) did not produce a significantly different amount of astaxanthin that any of their corresponding parental lines (Fig 2 and S2 Table). The pattern observed in males from crosses of southern populations was similar to the pattern of astaxanthin production observed in lines from the cross between southern SD and northern AB populations. In this cross, some RILs produced significantly less astaxanthin compared to non-hybrid controls, while other RILs did not (S5 Fig and S3 Table); however, overall, hybrid lines from this cross converted less astaxanthin than parental

lines (mean log astaxanthin ± SE; hybrid = 0.88 ± 0.04 ug mg; parental = 0.71 ± 0.06; n = 35; df = 15; $p$ = 0.044; S5 Fig and S3 Table).

The patterns in astaxanthin concentrations across RILs described above was mirrored when we repeated the analysis using the ratio of astaxanthin: dietary carotenoids (S6 Fig and S4 Table). We consistently detected two dietary carotenoids in our yeast-fed inbred line samples: β-carotene and hydroxyechinenone (S7 Fig). Both are precursors to astaxanthin found in *T. californicus* fed *Tetraselmis* algae [34]. Hydroxyechinenone is an intermediate between β-carotene and astaxanthin [9]. Hydroxyechinenone was the most abundant dietary carotenoid found in our yeast-fed inbred lines (p <0.001; S8 Fig). Dietary carotenoid concentrations did not significantly vary among yeast-fed inbred line samples (S9 Fig and S5 Table).

Although the data from yeast-fed lines shows no clear breakdown of mitonuclear function in RILs compared to PILs, we found significant relationships between astaxanthin bioconversion and energy production across all lines. We found a statistically significant, negative relationship between astaxanthin production and Complex I ATP production ($\beta_1$ = -0.08 μg astaxanthin mg$^{-1}$; $p$ = 0.020; $R^2_{adj}$ = 0.44; $n$ = 13) (Fig 3A), but no clear relationship between astaxanthin production and Complex II ATP production ($\beta_1$ = -0.03 μg astaxanthin mg$^{-1}$; $p$ = 0.610; $R^2_{adj}$ = 0.52; $n$ = 12) (Fig 3B). We did not find a statistically significant relationship between astaxanthin production and offspring development rate ($\beta_1$ = 0.05 μg astaxanthin mg$^{-1}$; $p$ = 0.200; $R^2_{adj}$ = 0.72; $n$ = 11) (Fig 3C). When we repeated these analyses using the ratio of astaxanthin to dietary carotenoids, we found that the relationship between astaxanthin: dietary carotenoids and Complex I ATP production was still negative but fell short of statistical significance (S10 Fig). However, the relationship between astaxanthin: dietary carotenoids and offspring development rate was significantly positive (S10 Fig).

Among all lines, the relationship between ATP production and offspring development was not statistically significant (Complex I: $\beta_1$ = -0.40 log nmol ATP min$^{-1}$; $p$ = 0.118; $R^2_{adj}$ = 0.323; $n$ = 10, and Complex II: $\beta_1$ = -0.24 log nmol ATP min$^{-1}$; $p$ = 0.445; $R^2_{adj}$ = 0.21; $n$ = 9) (S11A and S11B Fig). Unexpectedly, the relationship between ATP production and mitochondrial volume was not significant either (Complex I: $\beta_1$ = -0.01 log nmol ATP min$^{-1}$; $p$ = 0.963; $R^2_{adj}$ = 0.30; $n$ = 13, and Complex II: $\beta_1$ = 6.2e$^{-3}$ log nmol ATP min$^{-1}$; $p$ = 0.979; $R^2_{adj}$ = 0.46, $n$ = 13) (S11C and S11D Fig).

## The reciprocal cross between SD and SCN populations

We repeated our experiment using a fresh cross between SD and SCN populations, this time feeding copepods *Tetraselmis* algae one week prior to measuring ATP production and offspring development rate to mitigate potential confounding effects of the yeast-only diet.

We found no significant difference in astaxanthin production between parental SD and SCN copepods and copepods from the F1 and F2 generations (mean μg mg$^{-1}$ astaxanthin ± SE; F1 generation: parental = 0.14 ± 0.01; F1 hybrid = 0.16 ± 0.01; n = 52, df = 61, $p$ = 0.552; F2 generation: parental = 0.14 ± 0.01; F2 hybrid = 0.16 ± 0.02, n = 36, df = 60, $p$ = 0.870) (Fig 4). However, by the third generation, F3 hybrid copepods showed a statistically significant decrease in astaxanthin production when compared to parental copepods (mean μg mg$^{-1}$ astaxanthin ± SE; parental = 0.14 ± 0.01; F3 hybrid = 0.06 ± 0.03, n = 33, df = 60, $p$ = 0.014) (Fig 4). This decrease in astaxanthin by generation three was observed in both cross directions (SD x SCN and SCN x SD) (S12 Fig). Dietary carotenoids were observed in only a small subset of samples from this cross. We observed lutein, which is not a precursor to astaxanthin, and a peak putatively identified as echinenone/hydroxyechinenone (see S3 File).

We found that some relationships between astaxanthin production and fitness-based traits measured using individuals fed *Tetraselmis* algae did not match the relationships observed

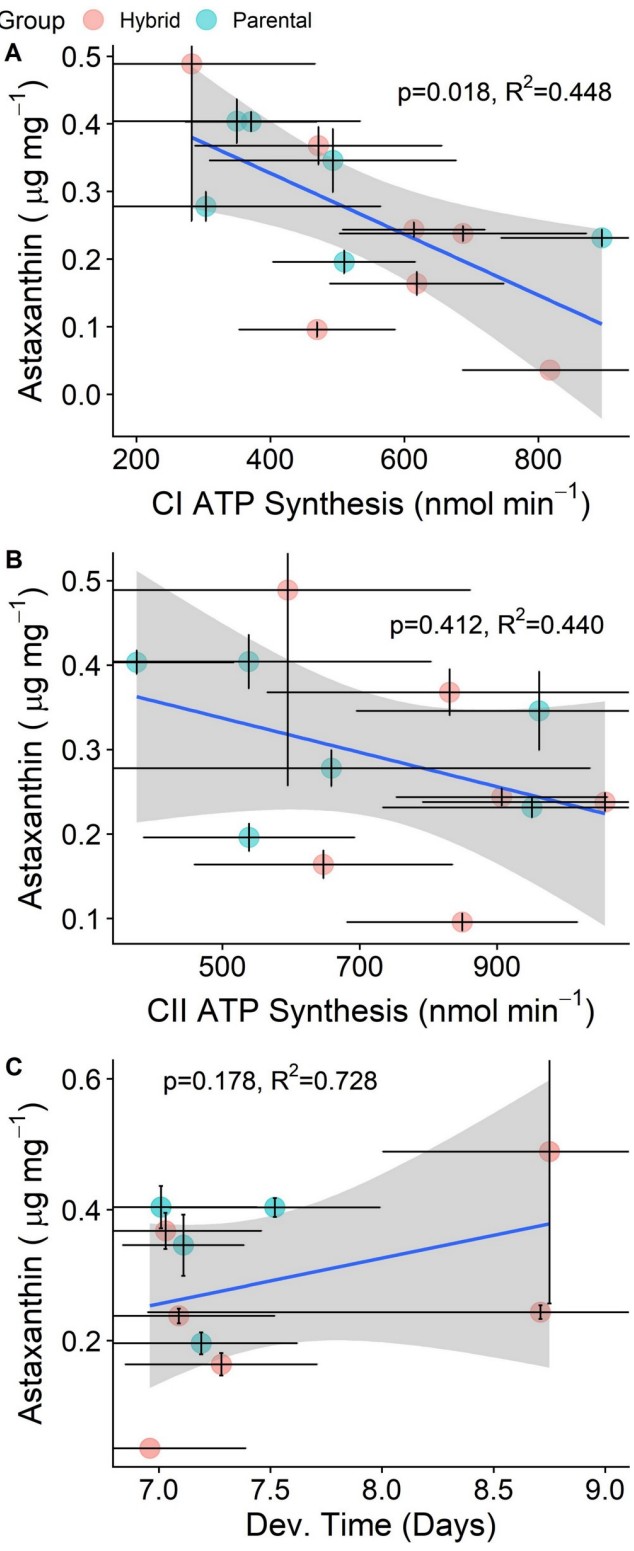

**Fig 3. Relationships between astaxanthin concentration and either ATP production or offspring development rate among all hybrid and non-hybrid lines.** Colored dots represent line averages and the vertical and horizontal black bars extending from the colored dots represent the standard error. The grey shading is the 95% confidence interval around the model estimated slope. Adjusted $R^2$ values are shown.

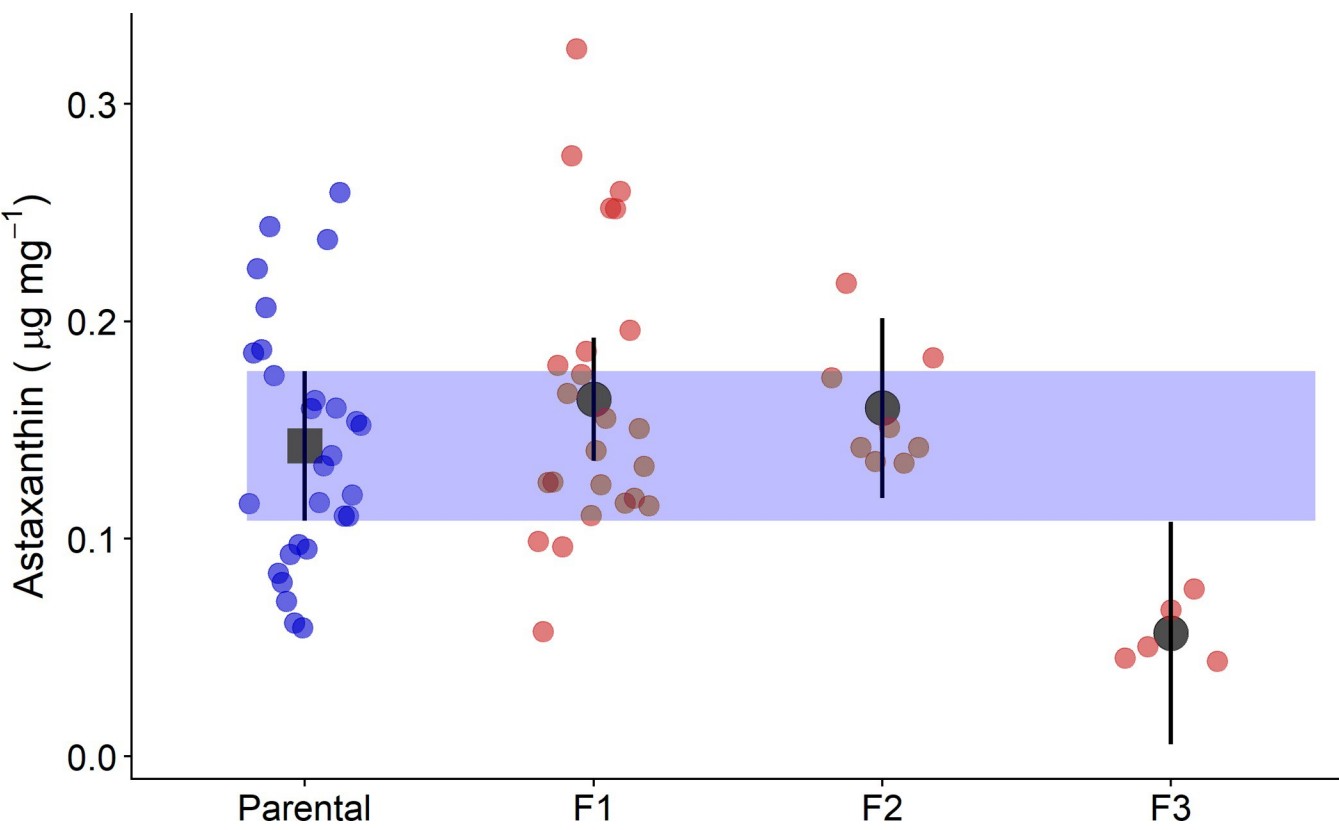

**Fig 4. Astaxanthin produced by copepods in each generation of a reciprocal cross between SD and SCN populations.** The parental group includes both SD and SCN copepods and each hybrid generation includes copepods from both cross directions (i.e., SD x SCN and SCN x SD. For the crosses split, see S5 Fig. Colored dots represent replicates within each line that are the average astaxanthin level of approximately 10 copepods. Dark grey circles show the generation average. Black lines and colored shading are 95% confidence intervals around the group average. Where confidence intervals do not overlap, there is a statistically significant difference between groups.

with the yeast-only inbred lines. The relationship between astaxanthin production and Complex I ATP production was positive this time, trending towards statistical significance ($\beta_1$ = 0.05 µg astaxanthin mg$^{-1}$; $p$ = 0.056; $R^2_{adj}$ = 0.84; $n$ = 4; Fig 5A), but there was no significant relationship between astaxanthin production and Complex II ATP production ($\beta_1$ = -0.03 µg astaxanthin mg$^{-1}$; $p$ = 0.388; $R^2_{adj}$ = 0.06; $n$ = 4; Fig 5B). The relationship between astaxanthin production and offspring development rate was negative and statistically significant ($\beta_1$ = -0.04 µg astaxanthin mg$^{-1}$; $p$ = 0.016; $R^2_{adj}$ = 0.59; $n$ = 8; Fig 5C).

Unlike with the multigeneration inbred lines fed yeast during fitness measurements, copepods from the SD x SCN crosses fed algae showed a positive, statistically significant relationship between Complex I ATP production and mitochondrial volume ($\beta_1$ = 0.98 log nmol CS min$^{-1}$; $p$ = 0.023; $R^2_{adj}$ = 0.93; $n$ = 4; S13A Fig), but there was no significant relationship between Complex II ATP production and mitochondrial volume ($\beta_1$ = 0.63 log nmol CS min$^{-1}$; $p$ = 0.371; $R^2_{adj}$ = 0.09; $n$ = 4; S13B Fig). The relationship between Complex I ATP and offspring development rate was negative, but not statistically significant ($\beta_1$ = -0.73 log days; $p$ = 0.087; $R^2_{adj}$ = 0.75; $n$ = 4; S13C Fig). There was no significant relationship between Complex II ATP production and offspring development rate ($\beta_1$ = 0.13 log days; $p$ = 0.837; $R^2_{adj}$ = -0.46; $n$ = 4; S13D Fig).

Comparing F3 hybrid lines and parental lines, there were no statistically significant differences in Complex I ATP production or Complex II ATP production. However, F3 hybrids did

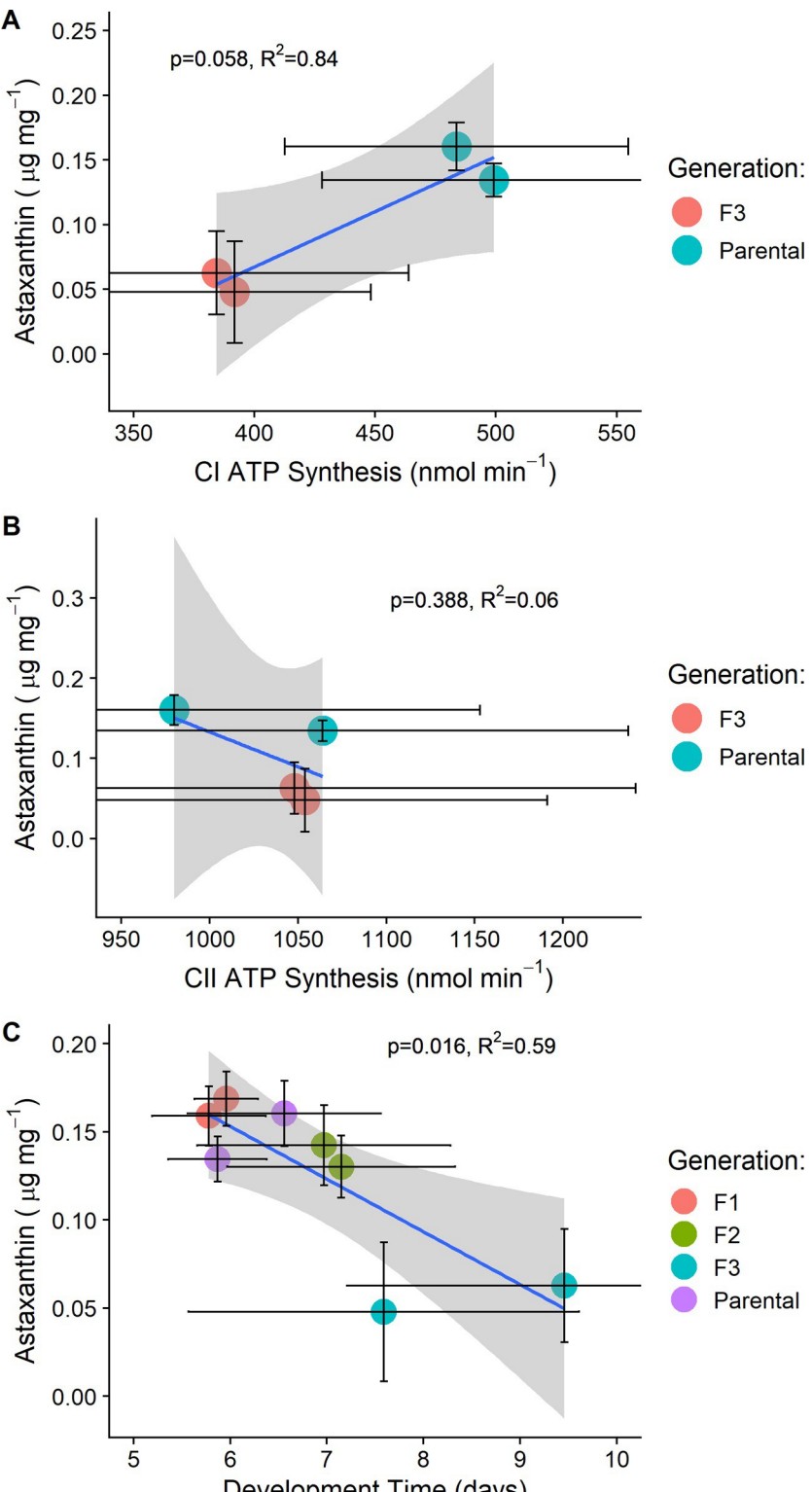

**Fig 5.** Relationships between astaxanthin production and either ATP production (panels A and B) and offspring development rate (panel C) among hybrid and non-hybrid lines from the cross between SD and SCN. Colored dots represent line averages and black bars extending from the colored dots represent the standard error of each trait on the x- and y-axis. The grey shading is the 95% confidence interval around the model estimated slope. Adjusted $R^2$ values are shown.

consistently make less Complex I ATP on average with reduced variation compared to parental lines (S14A and S14C Fig). This decrease was not observed in Complex II ATP production (S14B Fig). Offspring development rate increased progressively from the parental generation to the F3 hybrid generation. F3 hybrids had a statistically significant longer offspring development rate than parental copepods (mean log days ± SE; parental = 6.3 ± 0.56; F3 hybrid = 8.4 ± 0.49, n = 21, df = 43, *p* = 0.036) (S14C Fig) and F1 hybrid copepods (mean log days ± SE; F1 hybrid = 6.6 ± 0.0.42; F3 hybrid = 0.06 ± 0.03, n = 28, df = 43, *p* = 0.045) (S14C Fig).

## Discussion

Previous research demonstrated that interpopulation hybrid lines of *T. californicus* showed high variability but a consistent pattern of reduction in ATP production [43], ETS complex activity [42], oxidative homeostasis [40], fecundity [40, 41], offspring survivorship [43], and offspring development [43]. It was demonstrated that the observed decline of performance in hybrid copepods was likely due to mitochondrial dysfunction caused by mitonuclear incompatibilities brought on through the shuffling of nuclear alleles through sexual recombination [39, 59]. Thus, we predicted that if ketolation of yellow dietary carotenoids to red carotenoids is tied to mitochondrial function, then we would observe reduced astaxanthin content in hybrid copepods that also showed evidence of reduced mitochondrial performance and fitness. However, we observed no clear reduction in mitochondrial performance or offspring development in our hybrid RIL copepods compared to parental lines (S2 and S3 Figs). Yet, some RILs bioconverted significantly less astaxanthin compared to corresponding PILs, while others did not (Figs 2 and S5 and S6). This variation is to be expected since each RIL captures a unique set of parental alleles; on average, the expectation is that mitonuclear coadaptation will be disrupted. However, those lines with the most severe disruption will die (and not appear in our data set) while some RILs will obtain favorable mitonuclear combinations and continue to propagate. It is unclear if there is any way to rescue RILs with severely incompatible mitonuclear combinations that are in danger of extinction. Experiments have shown that backcrossing to the maternal population can reintroduce compatible nuclear alleles and may rescue viability (reviewed in [39]). However, there have also been experiments that show that mitonuclear incompatibilities differ among crosses [60–62]. Therefore, "fixes" for dying lines may not work equally in all cases. Moreover, fixes that alter the environment (for example, via diet or temperature changes) to relieve the stress of mitonuclear incompatibilities may only make it more difficult to compare fitness measures among lines.

We hypothesize that a key confounding factor in this study was dietary stress resulting from the poor yeast diet we used to produce copepods deficient of carotenoid pigments. Malnourishment has been observed to negatively impact oxidative state and mitochondrial function [63, 64]. We found that lines fed the carotenoid-devoid yeast diet produced less ATP (via Complex I substrates) than the same lines fed a complete, lipid and antioxidant rich diet of photosynthetic algae (S1 Fig). We also observed that offspring from yeast-fed parental lines developed significantly slower than offspring from parental lines fed algae (S1 Fig). Moreover, among lines fed yeast only, we found no significant relationship between ATP production and mitochondrial volume (S11C and S11D Fig). This is an important result because typically, we expect a positive relationship between ATP production and CS activity [65, 66]. Indeed, CS activity is used to standardize measurements of ATP production to account for variation in mitochondrial volume when sampling [56, 66]. Thus, it is possible that dietary restriction obscured the relationship between ATP production and CS activity. Whether or not this means a link between astaxanthin bioconversion and mitonuclear incompatibilities was artificially obscured is unclear, especially considering that we observed little to no variation in

female copepods from the yeast-fed multigeneration lines (S4 Fig). However, the difference that we observed between male and female copepods may agree with other evidence that female *T. californicus* are more resistant to stressful conditions [67–70]. It is also possible that males are less resistant to changes in oxidative stress [71]. Female *T. californicus* must sequester enough astaxanthin not only for themselves, but also for their developing offspring. Carotenoids must be transferred to the developing eggs in order to provide freshly hatched offspring with the UV-protection they need to survive their first two naupliar instars [9]. During this early naupliar period, offspring (which are the same bright red as adults) cannot eat [53] and, therefore, cannot acquire their own carotenoid precursors. Indeed, it is common for mothers to deposit carotenoids into developing eggs in oviparous taxa [72–74].

Switching diets appeared to alleviate problems with the initial experiment and yielded results that were more suggestive of hybrid breakdown (S13 and S14 Figs). When we fed hybrid and parental copepod lines algae one week prior to making mitochondrial measurements, the relationship between Complex I ATP production and mitochondrial volume was significant and positive (S13A Fig) across lines as expected from the perspective of mitochondrial physiology [65, 66]. We also found a more consistent pattern between hybrid and parental lines: hybrid animals made slightly less Complex I ATP (S14A Fig), and had offspring with significantly longer development rates by the third generation (S14C Fig). These results represent only one reciprocal cross, however, and it is unclear if these results would also be seen in crosses between other populations.

Despite a lack of observable mitonuclear dysfunction in RILs fed yeast, we found an interesting relationship between astaxanthin bioconversion and ATP production among these lines (Fig 3). After statistically controlling for the non-independence of data within hybrid vs. parental lines, we found a significant, negative relationship between Complex I ATP production and astaxanthin bioconversion (Fig 3A). We found no clear relationship between astaxanthin bioconversion and Complex II ATP production (Fig 3B). This negative relationship was also seen when ATP production was modeled against the ratio of astaxanthin: dietary carotenoids, although it fell short of statistical significance (S10 Fig). This difference is interesting for two reasons. First, unlike Complex I, Complex II is encoded solely by the nuclear genome [44, 75]. This suggests astaxanthin pigmentation may better predict Complex I function than Complex II function within a given line. In previous studies, Complex II was found to be the only ETS Complex not impaired by hybridization [42]. Likewise, we found a slight decrease in Complex I ATP production, but not in Complex II ATP production, in the SD x SCN reciprocal cross fed algae (S9A and S9B Fig). Second, Complex II utilizes a different electron donor and acceptor system than Complex I [76]. Complex I uses NADH as its primary electron donor, producing NAD+ as a result, while Complex II uses FADH as an electron donor, producing FAD+. It is possible that carotenoid ketolase enzymes, probably members of the Cytochrome P450 family or CrtS-like enzymes [18, 77], also operate using NADH/NAD + or NADPH/NADP+ systems [22, 26, 78, 79]. Perhaps the stronger relationship between astaxanthin bioconversion and Complex I ATP production could be explained by a shared pool of electron donors [79].

A shared pool of electron donors between Complex I and carotenoid ketolase enzymes may also explain why we observed a negative relationship between Complex I ATP production and astaxanthin bioconversion. It is possible that under stressful conditions, when demand for energy is high relative to available substrates for oxidative phosphorylation, NADH or NADPH used to catalyze reactions in Complex I will no longer be available to catalyze carotenoid ketolation, thus invoking trade-off between production of ATP or production of astaxanthin. However, under benign conditions when the demand for ATP is lower, the relationship could then be positive, as we observed with the SD x SCN reciprocal cross fed a

complete algae diet during fitness measurements. In molting house finches, males that bioconverted the most red ketocarotenoids also showed signs of withstanding higher levels of cellular stress in their mitochondria, as indicated by lower rates of mitochondrial replacement despite higher levels of oxidative damage to their cellular membranes [26]. This may help explain why we found no significant relationship between ATP production and mitochondrial volume in multigeneration inbred lines fed yeast only.

Alternatively, some invertebrates, including *T. californicus*, can also make use of a nuclear-encoded alternative oxidase (AOX) in their ETS to combat oxidative stress [80–82]. An increase in AOX expression would continue to consume oxygen for respiration, but would decrease ATP production due to reduced electron flow to Complex IV [81]. Similarly, the use of uncoupling proteins to reduce reactive oxygen species and maximize survival in the face of oxidative stress may also have resulted in a decrease in ATP [83–85]. If copepods in our study were using AOX or uncoupling enzymes to effectively combat oxidative stress due to hybridization and the poor yeast diet, it may explain why our most colorful lines also tended to produce less ATP (Fig 3). Certainly, ATP production can be informative in many situations, but it can also be an ambiguous metric when measured without the added context of metabolic rate [86]. Indeed, a deficiency in ATP can be compensated for through many different changes in mitochondrial dynamics [87–91]. Even though these ideas are largely speculative, a positive relationship between the metabolic rate and astaxanthin bioconversion has been observed in hybrid *T. californicus* [92]. In wild male house finches, individuals with greater energetic capacity are better able to bioconvert carotenoids [26]. Thus, carotenoid bioconversion may predict the capacity to meet changes in energy demand or withstand oxidative stress, rather than signal raw energy production.

The unexpected relationship between astaxanthin bioconversion and offspring development rate in multigeneration inbred lines (Figs 3 and S10) may also have been influenced by the limited yeast-only diet. Indeed, we observed that in the yeast-only lines, offspring development time was longer in lines with more efficient astaxanthin bioconversion (S10C Fig). Thus, paradoxically, in copepods fed yeast-only, lines with more astaxanthin produced less ATP and had slower developing offspring. However, this relationship was reversed in the reciprocal cross between SD and SCN when the dietary stress was relieved (Fig 5). In the yeast-only inbred lines, we found that offspring from all lines developed at similar rates (S3 Fig); however, in the reciprocal cross between SD and SCN fed algae, we found that development rate was significantly longer in hybrids by the third generation (S14C Fig). We have frequently observed longer development rates in laboratory cultures fed yeast-only diets, and in the data in this study we also observed that parental copepods fed yeast only produced offspring with longer development rates on average compared to parental copepods fed algae (S1E Fig). It could be that the malnourishing yeast diet limited variation in offspring development, slowing all lines and changing the relationship between offspring development and astaxanthin bioconversion. These results could support the idea that astaxanthin pigmentation is fundamentally tied to individual condition [2, 24], even in systems that do not experience sexual selection on carotenoid coloration [49].

Across diverse vertebrate [26, 93–95] and invertebrate species [37, 96, 97], coloration derived from the deposition of carotenoids is recognized as an honest signal of individual condition [2, 98–102]. For many years, it was assumed that carotenoid pigments were a limiting resource for many animals [102, 103]. It has been asserted that carotenoids perform physiological roles in body maintenance in many species, making them costly to devote toward pigmentation alone [104–106]. Carotenoids may be important signaling molecules in immune responses and may help combat oxidative stress through free radical scavenging or the stimulation of antioxidant activity [104, 107]. Previous hypotheses suggest there is a carotenoid-based resource trade-off between pigmentation and body maintenance [108, 109]. However,

the hypothesis that carotenoid coloration entails a costly trade-off has been called into question [24, 97, 110–112]. An alternative to the resource tradeoff hypothesis is the Shared-Pathway Hypothesis, which proposes that the bioconversion and accumulation of red carotenoids for coloration shares a biochemical pathway with mitochondrial metabolism in the cell [22, 24, 25, 113, 114]. The general prediction derived from this hypothesis is that ketocarotenoid bioconversion is an index that signals mitochondrial efficiency, flexibility, and capacity. Our results in *T. californicus*, may indicate that the Shared-Pathway Hypothesis is applicable in taxa beyond avian species, if indeed carotenoid bioconversion is an indicator of mitochondrial capacity to withstand stress. However, our results do not refute the idea that there is a tradeoff between body maintenance and carotenoid bioconversion in *T. californicus*. To better evaluate the hypotheses concerning a shared pathway between carotenoid bioconversion and mitochondrial function, specific measures of oxidative phosphorylation in the mitochondria, such as oxygen consumption, electron donor/acceptor ratios, and redox potential, may be important to consider in the future in order to contextualize energy production.

## Supporting information

**S1 Fig. ATP production and offspring development between algae-fed (Tetra.) and yeast-fed copepods.** Individual replicates from yeast-fed copepods are shown in red and data from algae-fed copepods are in blue. A) Difference in Complex I ATP production between the La Bufadora parental line on each diet, B) difference in Complex I ATP production between hybrid copepods from the BUF x SD 19 line on each diet, C) difference in Complex II ATP production from the La Bufadora parental line on each diet, D) difference in Complex II ATP production between the hybrid copepods from the BUF x SD 19 line on each diet, and E) difference in offspring development time between all parental copepod lines fed either algae or yeast. Black dots are group averages and black bars are 95% confidence intervals around the mean.
(TIF)

**S2 Fig.** Differences among hybrid RILs and non-hybrid PILs from multigeneration crosses fed yeast only in A-B) Complex I ATP production and C-D) Complex II ATP production. Colored dots represent replicates within each line that are the average ATP produced by isolated mitochondria from 20 pooled copepods. Dark grey circles are the line average. Black lines and colored shading are 95% confidence intervals around the line average. Where 95% confidence intervals do not overlap, there is a statistically significant difference between lines.
(TIF)

**S3 Fig. Differences among hybrid RILs and non-hybrid PILs from multigeneration crosses fed yeast only in offspring development time (days post hatch to first copepodid stage).** Colored dots represent replicates within each line that are the average values produced by individual females. Dark grey circles are the line average. Black lines and colored shading are 95% confidence intervals around the line average. Where 95% confidence intervals do not overlap, there is a statistically significant difference between lines.
(TIF)

**S4 Fig. Astaxanthin production among female hybrid RILs and non-hybrid PILs from multigeneration crosses.** Colored dots represent replicates within each line that are the average astaxanthin level of approximately 10 copepods. Dark grey circles (parental lines) and dark grey squares (hybrid lines) are the line average. Black lines and colored shading are 95% confidence intervals around the line average. Where 95% confidence intervals do not overlap, there is a statistically significant difference between lines.
(TIF)

**S5 Fig.** Line averages (black dots) and individual replicates (red dots) in astaxanthin production among AB x SD hybrid lines (A-H; 'Hybrid' represents average among all lines) and in parental (AB) copepods. Black bars show standard error. After one generation of inbreeding, 9 AB parental lines and 8 AB x SD hybrid lines were viable, while SD parental lines and SD x AB hybrid lines did not survive to carotenoid analysis. Previously, *T. californicus* has been shown to be highly sensitive to inbreeding [115], which is likely the cause of the mortality that we observed in SD parental and SD x AB hybrid lines.
(TIF)

**S6 Fig. Differences in astaxanthin: Dietary carotenoid concentration ratio among hybrid RILs and non-hybrid PILs from multigeneration crosses.** Colored dots represent replicates within each line that are the average carotenoid ratio of approximately 10 copepods. Dark grey circles (parental lines) and dark grey squares (hybrid lines) are the line average. Black lines and colored shading are 95% confidence intervals around the line average. Where 95% confidence intervals do not overlap, there is a statistically significant difference between lines (S4 Table).
(TIF)

**S7 Fig.** A) HPLC chromatograms of a mix of carotenoid standards (blue), hydroxyechinenone standard (red), a copepod sample with a low ratio of astaxanthin to dietary carotenoids (black), and a copepod samples with a high ratio of astaxanthin to dietary carotenoids (yellow). Numbered peaks are as follows: 1) astaxanthin, 2) lutein, 3) canthaxanthin, 4) hydroxyechinenone, 5) β-carotene. B) HPLC chroma profile of the Tetraselmis algae we fed the copepods during the experiment. Numbered peaks are as follows: 1) violaxanthin isomer, 2) trans-violaxanthin, 3) lutein, 4) hydroxyechinenone, 5) α-carotene, 6) β-carotene. Of these carotenoids, only trans-violaxanthin, hydroxyechinenone and β-carotene are precursors to astaxanthin [9, 32].
(TIF)

**S8 Fig. Ridgeline plot showing the distribution of hydroxyechinenone and β-carotene concentrations across all copepod samples from the yeast-only RILs and PILs.** The black lines represent the median and upper/lower quartiles of each group. The p-value was derived from a linear mixed effects model with inbred line ID encoded as a random effect.
(TIF)

**S9 Fig. Differences in dietary carotenoid concentration among hybrid RILs and non-hybrid PILs from multigeneration crosses.** Colored dots represent replicates within each line that are the average of approximately 10 copepods. Dark grey circles (parental lines) and dark grey squares (hybrid lines) are the line average. Black lines and colored shading are 95% confidence intervals around the line average. Where 95% confidence intervals do not overlap, there is a statistically significant difference between lines (S5 Table).
(TIF)

**S10 Fig. Relationships between astaxanthin: Dietary carotenoids and either ATP production or offspring development rate among all hybrid and non-hybrid lines.** Colored dots represent line averages and the vertical and horizontal black bars extending from the colored dots represent the standard error of each trait. The grey shading is the 95% confidence interval around the model estimated slope. Adjusted $R^2$ values are shown.
(TIF)

**S11 Fig.** Relationships between ATP production and either offspring development time (days post hatch to first copepodid stage; A and B) or mitochondrial volume (C and D) among

hybrid and non-hybrid lines. Colored dots represent line averages and the vertical and horizontal black bars extending from the colored dots represent the standard error. The grey shading is the 95% confidence interval around the model estimated slope. Adjusted $R^2$ values are shown.
(TIF)

**S12 Fig. Astaxanthin produced by copepods each generation of a reciprocal cross between SD and SCN populations.** Groups are split by the direction of the cross (SCN female x SD male on the left, and SD female x SCN male on the right). Colored dots represent replicates within each group that are the average astaxanthin level of approximately 10 copepods per replicate. Dark grey circles are the group average. Black lines and colored shading are 95% confidence intervals around the group average. Where confidence intervals do not overlap, there is a statistically significant difference between groups.
(TIF)

**S13 Fig.** Relationships between ATP production and mitochondrial volume (A and B) or offspring development time (C and D) among hybrid and non-hybrid lines from the reciprocal cross between SD and SCN populations. Colored dots represent line averages and the vertical and horizontal black bars extending from the colored dots represent the standard error. The grey shading is the 95% confidence interval around the model estimated slope. Adjusted $R^2$ values are shown.
(TIF)

**S14 Fig.** Differences among hybrid RILs and non-hybrid PILs from the reciprocal cross in A) Complex I ATP production, B) Complex II ATP production, and C) offspring development rate. Colored dots represent replicates within each line that are either the average ATP produced by isolated mitochondria from 20 pooled copepods (A and B) or clutch averages from individual females (C). Large, dark grey circles and squares are the line average. Black lines are 95% confidence intervals around the line average. Where 95% confidence intervals do not overlap, there is a statistically significant difference between lines.
(TIF)

**S1 Table. A list of the lines from the experimental crosses used in this study.** The number of the replicates per line for each measurement and the approximate number of individual copepods per replicate. Shown as: # of line replicates (approx. # of individuals per replicate).
(DOCX)

**S2 Table. Results from statistical models of paired contrasts of astaxanthin production among male copepods from multigeneration RILs and PILs.** The model beta estimate represents the difference in the means per group. The confidence limits in the right two columns represent the confidence boundaries around the model estimate. Bold lines denote a significant difference between the contrast and reference groups.
(DOCX)

**S3 Table. Results from statistical models of paired contrasts of astaxanthin production among copepods from preliminary RILs from the cross between SD and AB populations.** The model beta estimate represents the difference in the means per group. The confidence limits in the right two columns represent the confidence boundaries around the model estimate.
(DOCX)

**S4 Table. Results from statistical models of paired contrasts of astaxanthin: Dietary carotenoid concentration ratio among male copepods from multigeneration RILs and PILs.** The

model beta estimate represents the difference in the means per group. The confidence limits in the right two columns represent the confidence boundaries around the model estimate. Bold lines denote a significant difference between the contrast and reference groups.
(DOCX)

**S5 Table. Results from statistical models of paired contrasts of dietary carotenoid accumulation among male copepods from multigeneration RILs and PILs.** The model beta estimate represents the difference in the means per group. The confidence limits in the right two columns represent the confidence boundaries around the model estimate. Bold lines denote a significant difference between the contrast and reference groups.
(DOCX)

**S1 File. Supplemental methods with protocols and further details.**
(DOCX)

**S2 File. The R code used to analyze data.**
(TXT)

**S3 File. All datasets in a master worksheet.**
(XLSX)

## Acknowledgments

We would like to thank Dr. Timothy Healy, Dr. Paul Cobine, Reggie Blackwell, and Elliot Weiss for their assistance in the completion of this study. We would also like to thank reviewers, Dr. Emily Webb and another who chose to remain anonymous, for comments that improved this manuscript.

## Author Contributions

**Conceptualization:** Ronald S. Burton, Geoffrey E. Hill, Ryan J. Weaver.

**Data curation:** Matthew J. Powers, Lucas D. Martz, Ryan J. Weaver.

**Formal analysis:** Matthew J. Powers, Ryan J. Weaver.

**Funding acquisition:** Ronald S. Burton, Geoffrey E. Hill, Ryan J. Weaver.

**Investigation:** Matthew J. Powers, Lucas D. Martz, Ryan J. Weaver.

**Methodology:** Matthew J. Powers, Lucas D. Martz, Ryan J. Weaver.

**Project administration:** Ronald S. Burton, Geoffrey E. Hill.

**Resources:** Ronald S. Burton, Geoffrey E. Hill.

**Supervision:** Matthew J. Powers, Ronald S. Burton, Geoffrey E. Hill.

**Validation:** Matthew J. Powers, Lucas D. Martz, Ryan J. Weaver.

**Visualization:** Matthew J. Powers, Ryan J. Weaver.

**Writing – original draft:** Matthew J. Powers.

**Writing – review & editing:** Matthew J. Powers, Lucas D. Martz, Ronald S. Burton, Geoffrey E. Hill, Ryan J. Weaver.

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
