## [Decision Letter · Decision Letter 0]

31 Aug 2021

PONE-D-21-20778

Evidence for hybrid breakdown in production of red carotenoids in a marine invertebrate Tigriopus californicus

PLOS ONE

Dear Dr. Powers,

Thank you for submitting your manuscript to PLOS ONE. After careful consideration, we feel that it has merit but does not fully meet PLOS ONE’s publication criteria as it currently stands. Therefore, we invite you to submit a revised version of the manuscript that addresses the points raised during the review process.

Your manuscript received two thoughtful, constructive reviews by experts in the field. They both find the paper of high quality and interest, but they also raise several issues that need to be addressed before the manuscript can be published. In particular, one reviewer thinks that a weakness can be addressed by possibly redoing some of the analyses to account for variation in dietary carotenoids that have been absorbed, but not converted into astaxanthin.  There also many other comments and suggestions from both reviewers, but I suspect that can all be addressed. 

We look forward to receiving your revised manuscript.

Kind regards,

Hans G. Dam, Ph. D.

Academic Editor

PLOS ONE

2. In your Methods section, please provide additional information regarding the permits you obtained for the collection of copepod samples. Please ensure you have included the full name of the authority that approved the field site access and, if no permits were required, a brief statement explaining why.

Reviewers' comments:

Reviewer's Responses to Questions

**Comments to the Author**

1. Is the manuscript technically sound, and do the data support the conclusions?

Reviewer #1: Yes

Reviewer #2: Yes

2. Has the statistical analysis been performed appropriately and rigorously? 

Reviewer #1: Yes

Reviewer #2: Yes

3. Have the authors made all data underlying the findings in their manuscript fully available?

Reviewer #1: Yes

Reviewer #2: Yes

4. Is the manuscript presented in an intelligible fashion and written in standard English?

Reviewer #1: Yes

Reviewer #2: Yes

5. Review Comments to the Author

Reviewer #1: The authors investigate whether production of the red keto-carotenoid astaxanthin from dietary precursors is disrupted in inbred hybrid lines (RIL’s) generated from crosses between populations of the marine copepod Tigriopus californicus in which incompatibilities between nuclear and mitochondrial genes involved in the process could be exposed. Production of astaxanthin is an enzymatic process that has been hypothesized to signal individual condition in some organisms, possibly through shared metabolic pathways involved in oxidative phosphorylation.

The authors do observe reduced capabilities to produce astaxanthin in males (why not females?!) of some of the created inbred lines compared to the parental forms that may be attributed to an effect of the genetic background, but little in the way of a predicted link to mitochondrial performance. Also, the results are muddied by an unexpected effect of diet (yeast-based without carotenoids vs algae-based) used to propagate the lines, such that some of the results, e.g., how astaxanthin production is related to ATP production, went in opposite directions between lines grown on the two different diets.

Personally, I feel that the study raises more questions than it answers. Fortunately, providing a bit more background information on the system studied may alleviate some of these deficiencies.

How many chromosomes or linkage groups does Tigriopus have? Of course, it would have been interesting to identify which chromosomes or linkage groups are associated with the inbred lines with reduced pigmentation, but I can accept that this is a different study. What could have been done to prevent RIL’s with specific combinations of nuclear/mitochondrial genes resulting in low viability to be selected against or to go “extinct”? Also, it is unfortunate that the authors weren’t able to use the same inbred lines to compare the results for the two diets.

The authors do come up with reasonable explanations for their results, including those linked to diet, but without additional measures of mitochondrial function to constrain them, it is hard to judge whether they are correct. However, on whole I feel that the authors do provide a fairly balanced treatment of their findings, even if they are somewhat inconclusive. They certainly raise questions that could, and should, be addressed in future work, building on the present work.

Specific (very minor) comments:

Line 84: “crossbills” (no hyphenation).

Line 105: remove extra space between “population” and “[37, 41]”.

Line 113: Should “complexes” be pluralized?

Line 160: “established”

Line 200: a description of, at the very least, the mobile phase would be appreciated.

Between lines 244 and 257 μL (not uL).

Line 250: “Spinazzi, Casarin (52)” shouldn’t this be “Spinazzi et al. [52]”?

Line 282: I am not sure that the word “Recombinant” is the best (or even correct) term here. It conjures images of recombinant DNA, which involves a different process. But I understand what you mean.

Line 307-308: “hybrid breakdown” may be too strong a word here.

Line 405: remove period before “T. californicus“

Line 492: remove space before “[76]”

Line 699: 59 is out of place

Reviewer #2: The authors present a set of experiments looking at relationships between astaxanthin concentration, mitochondrial function, and fitness in copepods. The authors hypothesized that astaxanthin would be correlated with mitochondrial function. They also evaluated links between mitochondrial function and fitness. The data collected revealed some unexpected and interesting relationships between these three variables that will make a valuable contribution to the field. Overall, I feel that this manuscript is well-written and addresses a gap in the literature of linking fitness to these other two variables that have been studied previously (astaxanthin, mitochondrial function) in this system. I also think that the study design and data analyses are mostly appropriate for addressing the hypotheses (see the one exception in my notes).

There are multiple aspects of this manuscript that I liked including:

• Fitness is rarely measured in this context, and copepods are a great system for it, so that is a much appreciated contribution.

• I appreciate the inclusion of R code for reproducibility of data analysis.

• I also appreciate the honesty of results presentation by publishing all of the results, not just the second experiment, after the authors realized that the diet might have a confounding effect. This helps those working in the system to avoid the same mistake.

However, there are a few aspects that need additional attention or clarification:

1. The authors mentioned “efficiency of carotenoid ketolation” in their hypothesis (line 118), however, astaxanthin concentration is not necessarily a measure of this, because it ignores variation in dietary carotenoids (e.g. beta-carotene) that were not converted into astaxanthin. If dietary carotenoids were measured (as I assume they were since beta-carotene and other dietary carotenoid standard curves were mentioned in the methods), then they should be used to control for differences in absorption of dietary carotenoids. There are many ways this could be addressed by the authors that would be satisfying. Some options include: 1) providing some evidence that total carotenoids (dietary + astaxanthin) did not vary substantially in your sample, 2) using total carotenoids and/or dietary carotenoids to make a ratio (e.g. astaxanthin : beta-carotene) that more accurately reflects bioconversion (not just concentration of astaxanthin), or 3) including total carotenoids in the models. If the authors don’t have data on dietary carotenoids in the sample for whatever reason, then a citation on prior work that has looked at dietary vs. astaxanthin concentrations in copepods would be acceptable.

2. In general, the hybrid-making methods (experimental crosses and creation of recombinant inbred lines) are lacking sufficient justification, either in the form of citations or written explanation by the authors’ experiences. Some justification could address why specific numbers of generations were used to create the hybrids, why discrete generations were maintained until the seventh generation when offspring were allowed to mate continuously, why with a particular cross (SD x AB) astaxanthin bioconversion was only assayed at the 5th generation, etc. There are multiple instances like this where the how is thoroughly described but not the why. At least some should be in the main text, but most of this justification could go in the supplemental materials.

3. There are question marks like this “(???)” in the HPLC pigment analysis section of the supplemental files that should be removed.

4. In the methods, the authors mention that gravid females were fed powdered Spirulina instead of yeast. Is this the same algae diet that is mentioned throughout the manuscript or is this a different algae diet? This should be explicitly noted either way, since it is the first mention of the word Spirulina (the rest of the manuscript just refers to algae diet). If it is a different diet, then the nutritional differences between it and the other algae diet should be described and justified.

5. In the statistics section of methods (line 264), the authors describe “astaxanthin accumulation” which could be represented by multiple possible data types (e.g. amount, concentration, proportion). I didn’t see that it meant concentration specifically until I looked at Figure 2 and saw that the y-axis was concentration (ug/mg). The text should reference this more explicitly (that accumulation refers to concentration of astaxanthin in this study).

6. PLOS authors have the option to publish the peer review history of their article (what does this mean?). If published, this will include your full peer review and any attached files.

Reviewer #1: No

Reviewer #2: **Yes: **Emily Webb

---

## [Author Response · Author response to Decision Letter 0]

16 Sep 2021

Please see the Response to Reviewers document for a color-coded, comment-by-comment response to reviewer, journal, and editor comments.

---

## [Decision Letter · Decision Letter 1]

19 Oct 2021

Evidence for hybrid breakdown in production of red carotenoids in the marine invertebrate Tigriopus californicus

PONE-D-21-20778R1

Dear Dr. Powers,

We’re pleased to inform you that your manuscript has been judged scientifically suitable for publication and will be formally accepted for publication once it meets all outstanding technical requirements.

Kind regards,

Hans G. Dam, Ph. D.

Academic Editor

PLOS ONE

Additional Editor Comments (optional):

Reviewers' comments:

Reviewer's Responses to Questions

**Comments to the Author**

1. If the authors have adequately addressed your comments raised in a previous round of review and you feel that this manuscript is now acceptable for publication, you may indicate that here to bypass the “Comments to the Author” section, enter your conflict of interest statement in the “Confidential to Editor” section, and submit your "Accept" recommendation.

Reviewer #2: All comments have been addressed

2. Is the manuscript technically sound, and do the data support the conclusions?

Reviewer #2: Yes

3. Has the statistical analysis been performed appropriately and rigorously? 

Reviewer #2: Yes

4. Have the authors made all data underlying the findings in their manuscript fully available?

Reviewer #2: Yes

5. Is the manuscript presented in an intelligible fashion and written in standard English?

Reviewer #2: Yes

6. Review Comments to the Author

Reviewer #2: The authors have appropriately and thoroughly addressed my comments, so I recommend that it be accepted for publication.

7. PLOS authors have the option to publish the peer review history of their article (what does this mean?). If published, this will include your full peer review and any attached files.

Reviewer #2: **Yes: **Emily Webb

---

## [Editor Report · Acceptance letter]

21 Oct 2021

PONE-D-21-20778R1 

Evidence for hybrid breakdown in production of red carotenoids in the marine invertebrate *Tigriopus californicus*

Dear Dr. Powers:

I'm pleased to inform you that your manuscript has been deemed suitable for publication in PLOS ONE. Congratulations! Your manuscript is now with our production department. 

Kind regards, 

on behalf of

Dr. Hans G. Dam 

Academic Editor

PLOS ONE